# Deep Neural Nets with Interpolating Function as Output Activation

**Bao Wang**
Department of Mathematics
University of California, Los Angeles
wangbaonj@gmail.com

Xiyang Luo
Department of Mathematics
University of California, Los Angeles
xylmath@gmail.com

Zhen Li
Department of Mathematics
HKUST, Hong Kong
lishen03@gmail.com

Wei Zhu
Department of Mathematics
Duke University
zhu@math.duke.edu

Zuoqiang Shi
Department of Mathematics
Tsinghua University
zqshi@mail.tsinghua.edu.cn

Stanley J. Osher
Department of Mathematics
University of California, Los Angeles
sjo@math.ucla.edu

## Abstract

We replace the output layer of deep neural nets, typically the softmax function, by a novel interpolating function. And we propose end-to-end training and testing algorithms for this new architecture. Compared to classical neural nets with softmax function as output activation, the surrogate with interpolating function as output activation combines advantages of both deep and manifold learning. The new framework demonstrates the following major advantages: First, it is better applicable to the case with insufficient training data. Second, it significantly improves the generalization accuracy on a wide variety of networks. The algorithm is implemented in PyTorch, and the code is available at https://github.com/BaoWangMath/DNN-DataDependentActivation.

## 1 Introduction

Generalizability is crucial to deep learning, and many efforts have been made to improve the training and generalization accuracy of deep neural nets (DNNs) [3, 14]. Advances in network architectures such as VGG networks [28], deep residual networks (ResNets)[12, 13] and more recently DenseNets [16] and many others [6], together with powerful hardware make the training of very deep networks with good generalization capabilities possible. Effective regularization techniques such as dropout and maxout [15, 30, 10], as well as data augmentation methods [19, 28, 32] have also explicitly improved generalization for DNNs.

A key component of neural nets is the activation function. Improvements in designing of activation functions such as the rectified linear unit (ReLU) [8], have led to huge improvements in performance in computer vision tasks [23, 19]. More recently, activation functions adaptively trained to the data such as the adaptive piecewise linear unit (APLU) [1] and parametric rectified linear unit (PReLU) [11] have lead to further improvements in performance of DNNs. For output activation, support vector machine (SVM) has also been successfuly applied in place of softmax[29]. Though training DNNs with softmax or SVM as output activation is effective in many tasks, it is possible that alternative activations that consider manifold structure of data by interpolating the output based on both training and testing data can boost performance of the network. In particular, ResNets can be reformulated as solving control problems of a class of transport equations in the continuum limit [21, 5]. Transport

theory suggests that by using an interpolating function that interpolates terminal values from initial values can dramatically simplify the control problem compared to an ad-hoc choice. This further suggests that a fixed and data-agnostic activation for the output layer may be suboptimal.

To this end, based on the ideas from manifold learning, we propose a novel output layer named weighted nonlocal Laplacian (WNLL) layer for DNNs. The resulted DNNs achieve better generalization and are more robust for problems with a small number of training examples. On CIFAR10/CIFAR100, we achieve on average a 30%/20% reduction in terms of test error on a wide variety of networks. These include VGGs, ResNets, and pre-activated ResNets. The performance boost is even more pronounced when the model is trained on a random subset of CIFAR with a low number of training examples. We also present an efficient algorithm to train the WNLL layer via an auxiliary network. Theoretical motivation for the WNLL layer is also given from the viewpoint of both game theory and terminal value problems for transport equations.

This paper is structured as follows: In Section 2, we introduce the motivation and practice of using the WNLL interpolating function in DNNs. In Section 2.2, we explain in detail the algorithms for training and testing DNNs with WNLL as output layer. Section 3 provides insight of using an interpolating function as output layer from the angle of terminal value problems of transport equations and game theory. Section 4 demonstrates the effectiveness of our method on a variety of numerical examples.

## 2 Network Architecture

In coarse grained representation, training and testing DNNs with softmax layer as output are illustrated in Fig. 1 (a) and (b), respectively. In $k$th iteration of training, given a mini-batch training data $(\mathbf{X}, \mathbf{Y})$, we perform:

*Forward propagation:* Transform $\mathbf{X}$ into deep features by DNN block (ensemble of conv layers, nonlinearities and others), and then activated by softmax function to obtain the predicted labels $\tilde{\mathbf{Y}}$:

$$\tilde{\mathbf{Y}} = \text{Softmax}(\text{DNN}(\mathbf{X}, \Theta^{k-1}), \mathbf{W}^{k-1}).$$

Then compute loss (e.g., cross entropy) between $\mathbf{Y}$ and $\tilde{\mathbf{Y}}$: $\mathcal{L} = \text{Loss}(\mathbf{Y}, \tilde{\mathbf{Y}})$.

*Backpropagation:* Update weights $(\Theta^{k-1}, \mathbf{W}^{k-1})$ by gradient descent (learning rate $\gamma$):

$$\mathbf{W}^k = \mathbf{W}^{k-1} - \gamma \frac{\partial \mathcal{L}}{\partial \tilde{\mathbf{Y}}} \cdot \frac{\partial \tilde{\mathbf{Y}}}{\partial \mathbf{W}}, \quad \Theta^k = \Theta^{k-1} - \gamma \frac{\partial \mathcal{L}}{\partial \tilde{\mathbf{Y}}} \cdot \frac{\partial \tilde{\mathbf{Y}}}{\partial \tilde{\mathbf{X}}} \cdot \frac{\partial \tilde{\mathbf{X}}}{\partial \Theta}.$$

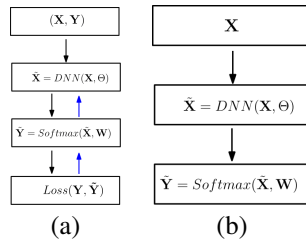

Figure 1: Training (a) and testing (b) procedures of DNNs with softmax as output activation layer.

Once the model is optimized, for testing data $\mathbf{X}$, the predicted labels are:

$$\tilde{\mathbf{Y}} = \text{Softmax}(\text{DNN}(\mathbf{X}, \Theta), \mathbf{W}),$$

for notational simplicity, we still denote the test set and optimized weights as $\mathbf{X}$, $\Theta$, and $\mathbf{W}$, respectively. In essence the softmax layer acts as a linear model on the space of deep features $\tilde{\mathbf{X}}$, which does not take into consideration the underlying manifold structure of $\tilde{\mathbf{X}}$. The WNLL interpolating function, which will be introduced in the following subsection, is an approach to alleviate this deficiency. Moreover, WNLL interpolation is based on the harmonic extension which avoids the curse of dimensionality issue in high dimensional interpolation.

## 2.1 Manifold Interpolation - An Harmonic Extension Approach

Let $\mathbf{X} = \{\mathbf{x}_1, \mathbf{x}_2, \cdots, \mathbf{x}_n\}$ be a set of points in a high dimensional manifold $\mathcal{M} \subset \mathbb{R}^d$ and $\mathbf{X}^{\text{te}} = \{\mathbf{x}_1^{\text{te}}, \mathbf{x}_2^{\text{te}}, \cdots, \mathbf{x}_m^{\text{te}}\}$ be a subset of $\mathbf{X}$. Suppose we have a (possibly vector valued) label function $g(\mathbf{x})$ defined on $\mathbf{X}^{\text{te}}$, and we want to interpolate a function $u$ that is defined on the entire manifold and can be used to label the entire dataset $\mathbf{X}$. Interpolation by using basis function in high dimensional space suffers from the curse of dimensionality. Instead, an harmonic extension is a natural and elegant approach to find such an interpolating function, which is defined by minimizing the following Dirichlet energy functional:

$$\mathcal{E}(u) = \frac{1}{2} \sum_{\mathbf{x}, \mathbf{y} \in \mathbf{X}} w(\mathbf{x}, \mathbf{y}) \left(u(\mathbf{x}) - u(\mathbf{y})\right)^2, \tag{1}$$

with the boundary condition:

$$u(\mathbf{x}) = g(\mathbf{x}), \ \mathbf{x} \in \mathbf{X}^{\text{te}},$$

where $w(\mathbf{x}, \mathbf{y})$ is a weight function, typically chosen to be Gaussian: $w(\mathbf{x}, \mathbf{y}) = \exp(-\frac{||\mathbf{x}-\mathbf{y}||^2}{\sigma^2})$ with $\sigma$ a scaling parameter. The Euler-Lagrange equation for Eq.(1) is:

$$\begin{cases} \sum_{\mathbf{y} \in \mathbf{X}} \left(w(\mathbf{x}, \mathbf{y}) + w(\mathbf{y}, \mathbf{x})\right) \left(u(\mathbf{x}) - u(\mathbf{y})\right) = 0 & \mathbf{x} \in \mathbf{X}/\mathbf{X}^{\text{te}} \\ u(\mathbf{x}) = g(\mathbf{x}) & \mathbf{x} \in \mathbf{X}^{\text{te}}. \end{cases} \tag{2}$$

By solving the linear system Eq.(2), we get the interpolated labels $u(\mathbf{x})$ for unlabeled data $\mathbf{x} \in \mathbf{X}/\mathbf{X}^{\text{te}}$. This interpolation becomes invalid when labeled data is tiny, i.e., $|\mathbf{X}^{\text{te}}| \ll |\mathbf{X}/\mathbf{X}^{\text{te}}|$. There are two solutions to resolve this issue: one is to replace the 2-Laplacian in Eq.(1) by a $p$-Laplacian [4]; the other is to increase the weights of the labeled data in the Euler-Lagrange equation [27], which gives the following weighted nonlocal Laplacian (WNLL) interpolating function:

$$\begin{cases} \sum_{\mathbf{y} \in \mathbf{X}} \left(w(\mathbf{x}, \mathbf{y}) + w(\mathbf{y}, \mathbf{x})\right) \left(u(\mathbf{x}) - u(\mathbf{y})\right) + \\ \left(\frac{|\mathbf{X}|}{|\mathbf{X}^{\text{te}}|} - 1\right) \sum_{\mathbf{y} \in \mathbf{X}^{\text{te}}} w(\mathbf{y}, \mathbf{x}) \left(u(\mathbf{x}) - u(\mathbf{y})\right) = 0 & \mathbf{x} \in \mathbf{X}/\mathbf{X}^{\text{te}} \\ u(\mathbf{x}) = g(\mathbf{x}) & \mathbf{x} \in \mathbf{X}^{\text{te}}. \end{cases} \tag{3}$$

For notational simplicity, we name the solution $u(\mathbf{x})$ to Eq.3 as $\text{WNLL}(\mathbf{X}, \mathbf{X}^{\text{te}}, \mathbf{Y}^{\text{te}})$. For classification tasks, $g(\mathbf{x})$ is the one-hot labels for the example $\mathbf{x}$. To ensure accuracy of WNLL, the labeled data should cover all classes of data in $\mathbf{X}$. We give a necessary condition in Theorem 1.

**Theorem 1.** *Suppose we have a data pool formed by $N$ classes of data uniformly, with the number of instances of each class be sufficiently large. If we want all classes of data to be sampled at least once, on average at least $N\left(1 + \frac{1}{2} + \frac{1}{3} + \cdots + \frac{1}{N}\right)$ data is need to be sampled from the data pool. In this case, the number of data sampled, in expectation for each class, is $1 + \frac{1}{2} + \frac{1}{3} + \cdots + \frac{1}{N}$.*

## 2.2 WNLL Activated DNNs and Algorithms

In both training and testing of the WNLL activated DNNs, we need to reserve a small portion of data/label pairs denoted as $(\mathbf{X}^{\text{te}}, \mathbf{Y}^{\text{te}})$, to interpolate the label $Y$ for new data. We name $(\mathbf{X}^{\text{te}}, \mathbf{Y}^{\text{te}})$ as the preserved template. Directly replacing softmax by WNLL (Fig. 2(a)) has difficulties in back propagation, namely, the true gradient $\frac{\partial \mathcal{L}}{\partial \Theta}$ is difficult to compute since WNLL defines a very complex implicit function. Instead, to train WNLL activated DNNs, we propose a proxy via an auxiliary neural nets (Fig. 2(b)). On top of the original DNNs, we add a buffer block (a fully connected layer followed by a ReLU), and followed by two parallel layers, WNLL and the linear (fully connected) layers. The auxiliary DNNs can be trained by alternating between the following two steps (training DNNs with linear and WNLL activations, respectively):

**Train DNNs with linear activation:** Run $N_1$ steps of the following forward and back propagation, where in $k$th iteration, we have:

*Forward propagation:* The training data $\mathbf{X}$ is transformed, respectively, by DNN, Buffer and Linear blocks to the predicted labels $\tilde{\mathbf{Y}}$:

$$\tilde{\mathbf{Y}} = \text{Linear}(\text{Buffer}(\text{DNN}(\mathbf{X}, \Theta^{k-1}), \mathbf{W}_B^{k-1}), \mathbf{W}_L^{k-1}).$$

Then compute loss between the ground truth labels $\mathbf{Y}$ and predicted ones $\tilde{\mathbf{Y}}$, denoted as $\mathcal{L}^{\text{Linear}}$ (e.g., cross entropy loss, and the same as following $\mathcal{L}^{\text{WNLL}}$).

*Backpropagation:* Update weights $(\Theta^{k-1}, \mathbf{W}_B^{k-1}, \mathbf{W}_L^{k-1})$ by gradient descent:

$$\mathbf{W}_L^k = \mathbf{W}_L^{k-1} - \gamma \frac{\partial \mathcal{L}^{\text{Linear}}}{\partial \tilde{\mathbf{Y}}} \cdot \frac{\partial \tilde{\mathbf{Y}}}{\partial \mathbf{W}_L}, \quad \mathbf{W}_B^k = \mathbf{W}_B^{k-1} - \gamma \frac{\partial \mathcal{L}^{\text{Linear}}}{\partial \tilde{\mathbf{Y}}} \cdot \frac{\partial \tilde{\mathbf{Y}}}{\partial \hat{\mathbf{X}}} \cdot \frac{\partial \hat{\mathbf{X}}}{\partial \mathbf{W}_B},$$

$$\Theta^k = \Theta^{k-1} - \gamma \frac{\partial \mathcal{L}^{\text{Linear}}}{\partial \tilde{\mathbf{Y}}} \cdot \frac{\partial \tilde{\mathbf{Y}}}{\partial \hat{\mathbf{X}}} \cdot \frac{\partial \hat{\mathbf{X}}}{\partial \tilde{\mathbf{X}}} \cdot \frac{\partial \tilde{\mathbf{X}}}{\partial \Theta}.$$

**Train DNNs with WNLL activation:** Run $N_2$ steps of the following forward and back propagation, where in $k$th iteration, we have:

*Forward propagation:* The training data $\mathbf{X}$, template $\mathbf{X}^{\text{te}}$ and $\mathbf{Y}^{\text{te}}$ are transformed, respectively, by DNN, Buffer, and WNLL blocks to get predicted labels $\hat{\mathbf{Y}}$:

$$\hat{\mathbf{Y}} = \text{WNLL}(\text{Buffer}(\text{DNN}(\mathbf{X}, \Theta^{k-1}), \mathbf{W}_B^{k-1}), \hat{\mathbf{X}}^{\text{te}}, \mathbf{Y}^{\text{te}}).$$

Then compute loss, $\mathcal{L}^{\text{WNLL}}$, between the ground truth labels $\mathbf{Y}$ and predicted ones $\hat{\mathbf{Y}}$.

*Backpropagation:* Update weights $\mathbf{W}_B^{k-1}$ only, $\mathbf{W}_L^{k-1}$ and $\Theta^{k-1}$ will be tuned in the next iteration in training DNNs with linear activation, by gradient descent.

$$\mathbf{W}_B^k = \mathbf{W}_B^{k-1} - \gamma \frac{\partial \mathcal{L}^{\text{WNLL}}}{\partial \hat{\mathbf{Y}}} \cdot \frac{\partial \hat{\mathbf{Y}}}{\partial \hat{\mathbf{X}}} \cdot \frac{\partial \hat{\mathbf{X}}}{\partial \mathbf{W}_B} \approx \mathbf{W}_B^{k-1} - \gamma \frac{\partial \mathcal{L}^{\text{Linear}}}{\partial \tilde{\mathbf{Y}}} \cdot \frac{\partial \tilde{\mathbf{Y}}}{\partial \hat{\mathbf{X}}} \cdot \frac{\partial \hat{\mathbf{X}}}{\partial \mathbf{W}_B}. \quad (4)$$

Here we use the computational graph of the left branch (linear layer) to retrieval the approximated gradients for WNLL. For a given loss value of $\mathcal{L}^{\text{WNLL}}$, we adopt the approximation $\frac{\partial \mathcal{L}^{\text{WNLL}}}{\partial \hat{\mathbf{Y}}} \cdot \frac{\partial \hat{\mathbf{Y}}}{\partial \hat{\mathbf{X}}} \approx \frac{\partial \mathcal{L}^{\text{Linear}}}{\partial \tilde{\mathbf{Y}}} \cdot \frac{\partial \tilde{\mathbf{Y}}}{\partial \hat{\mathbf{X}}}$ where the right hand side is also evaluated at this value. The main heuristic behind this approximation is the following: WNLL defines a harmonic function implicitly, and a linear function is the simplest nontrivial explicit harmonic function. Empirically, we observe this simple approximation works well in training the network. The reason why we freeze the network in the DNN block is mainly due to stability concerns.

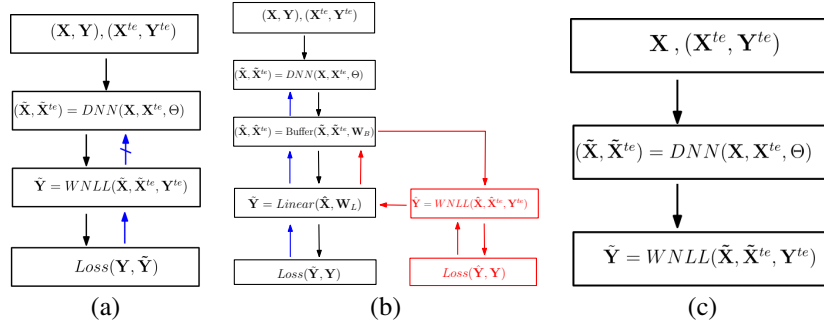

Figure 2: Training and testing procedure of the deep neural nets with WNLL as the last activation layer.(a): Direct replacement of the softmax by WNLL, (b): An alternating training procedure. (c): Testing.

The above alternating scheme is an algorithm of a greedy fashion. During training, WNLL activation plays two roles: on one hand, the alternating between linear and WNLL activations benefits each other which enables the neural nets to learn features that is appropriate for both linear classification and WNLL based manifold interpolation. On the other hand, in the case where we lack sufficient training data, the training of DNNs usually gets stuck at some bad local minima which cannot generalize well on new data. We use WNLL interpolation which provides a perturbation to the trained sub-optimal weights and can help to arrive at a local minima with better generalizability. At test time, we remove the linear classifier from the neural nets and use the DNN block together with WNLL to predict new data (Fig. 2 (c)). The reason for using WNLL instead of a linear layer is because WNLL is superior

to the linear classifier and this superiority is preserved when applied to deep features (which will be shown in Section. 4). Moreover, WNLL utilizes both the learned DNNs and the preserved template at test time which seems to be more stable to perturbations on the input data.

We summarize the training and testing procedures for the WNLL activated DNNs in Algorithms 1 and 2, respectively. In each round of the alternating procedure i.e., each outer loop in Algorithm. 1, the entire training set $(\mathbf{X}, \mathbf{Y})$ is first used to train the DNNs with linear activation. We randomly separate a template, e.g., half of the entire data, from the training set which will be used to perform WNLL interpolation in training WNLL activated DNNs. In practice, for both training and testing, we use minibatches for both the template and the interpolated points when the entire dataset is too large. The final predicted labels are obtained by a majority voted across interpolation results from all the template minibatches.

**Remark 1.** *In Algorithm. 1, the WNLL interpolation is also performed in mini-batch manner (as shown in the inner iteration). Based on our experiments, this does not reduce the interpolation accuracy significantly.*

---

**Algorithm 1** DNNs with WNLL as Output Activation: Training Procedure.

---

   **Input:** Training set: (data, label) pairs $(\mathbf{X}, \mathbf{Y})$.
   **Output:** An optimized DNNs with WNLL as output activation, denoted as $\mathrm{DNN_{WNLL}}$.
   **for** iter $= 1, \ldots, N$ (where $N$ is the number of alternating steps.) **do**
      //Train the left branch: DNNs with linear activation.
      Train DNN + Linear blocks, and denote the learned model as $\mathrm{DNN_{Linear}}$.
      //Train the right branch: DNNs with WNLL activation.
      Split $(\mathbf{X}, \mathbf{Y})$ into training data and template, i.e., $(\mathbf{X}, \mathbf{Y}) \doteq (\mathbf{X}^{\mathrm{tr}}, \mathbf{Y}^{\mathrm{tr}}) \bigcup (\mathbf{X}^{\mathrm{te}}, \mathbf{Y}^{\mathrm{te}})$.
      Partition the training data into $M$ mini-batches, i.e., $(\mathbf{X}^{\mathrm{tr}}, \mathbf{Y}^{\mathrm{tr}}) = \bigcup_{i=1}^{M} (\mathbf{X}_i^{\mathrm{tr}}, \mathbf{Y}_i^{\mathrm{tr}})$.
      **for** $i = 1, 2, \cdots, M$ **do**
         Transform $\mathbf{X}_i^{\mathrm{tr}} \bigcup \mathbf{X}^{\mathrm{te}}$ by $\mathrm{DNN_{Linear}}$, i.e., $\tilde{\mathbf{X}}^{\mathrm{tr}} \bigcup \tilde{\mathbf{X}}^{\mathrm{te}} = \mathrm{DNN_{Linear}}(\mathbf{X}_i^{\mathrm{tr}} \bigcup \mathbf{X}^{\mathrm{te}})$.
         Apply WNLL (Eq.(3)) on $\{\tilde{\mathbf{X}}^{\mathrm{tr}} \bigcup \tilde{\mathbf{X}}^{\mathrm{te}}, \mathbf{Y}^{\mathrm{te}}\}$ to interpolate label $\tilde{\mathbf{Y}}^{\mathrm{tr}}$.
         Backpropagate the error between $\mathbf{Y}^{\mathrm{tr}}$ and $\hat{\mathbf{Y}}^{\mathrm{tr}}$ via Eq.(4) to update $\mathbf{W}_B$ only.

---

**Algorithm 2** DNNs with WNLL as Output Activation: Testing Procedure.

---

   **Input:** Testing data $\mathbf{X}$, template $(\mathbf{X}^{\mathrm{te}}, \mathbf{Y}^{\mathrm{te}})$. Optimized model $\mathrm{DNN_{WNLL}}$.
   **Output:** Predicted label $\tilde{\mathbf{Y}}$ for $\mathbf{X}$.
   Apply the DNN block of $\mathrm{DNN_{WNLL}}$ to $\mathbf{X} \bigcup \mathbf{X}^{\mathrm{te}}$ to get the representation $\tilde{\mathbf{X}} \bigcup \tilde{\mathbf{X}}^{\mathrm{te}}$.
   Apply WNLL (Eq.(3)) on $\{\tilde{\mathbf{X}} \bigcup \tilde{\mathbf{X}}^{\mathrm{te}}, \mathbf{Y}^{\mathrm{te}}\}$ to interpolate label $\tilde{\mathbf{Y}}$.

---

## 3 Theoretical Explanation

In training WNLL activated DNNs, the two output activation functions in the auxiliary networks are, in a sense, each competing to minimize its own objective where, in equilibrium, the neural nets can learn better features for both linear and interpolation-based activations. This in flavor is similar to generative adversarial nets (GAN) [9]. Another interpretation of our model is the following: As noted in [21], in the continuum limit, ResNet can be modeled as the following control problem for a transport equation:

$$\begin{cases} \frac{\partial u(\mathbf{x}, t)}{\partial t} + \mathbf{v}(\mathbf{x}, t) \cdot \nabla u(\mathbf{x}, t) = 0 & \mathbf{x} \in \mathbf{X}, t \geq 0 \\ u(\mathbf{x}, 1) = f(\mathbf{x}) & \mathbf{x} \in \mathbf{X}. \end{cases} \tag{5}$$

Here $u(\cdot, 0)$ is the input of the continuum version of ResNet, which maps the training data to the corresponding label. $f(\cdot)$ is the terminal value which analogous to the output activation function in ResNet which maps deep features to the predicted label. Training ResNet is equivalent to tuning $\mathbf{v}(\cdot, t)$, i.e., continuous version of the weights, s.t. the predicted label $f(\cdot)$ matches that of the training data. If $f(\cdot)$ is a harmonic extension of $u(\cdot, 0)$, the corresponding weights $v(\mathbf{x}, t)$ would be close to zero. This results in a simpler model and may generalize better from a model selection point of view.

## 4 Numerical Results

To validate the classification accuracy, efficiency and robustness of the proposed framework, we test the new architecture and algorithm on CIFAR10, CIFAR100 [18], MNIST[20] and SVHN datasets [24]. In all experiments, we apply standard data augmentation that is widely used for the CIFAR datasets [12, 16, 31]. For MNIST and SVHN, we use the raw data without any augmentation. We implement our algorithm on the PyTorch platform [26]. All computations are carried out on a machine with a single Nvidia Titan Xp graphics card.

Before diving into the performance of DNNs with different output activation functions, we first compare the performance of WNLL with softmax on the raw input images for various datasets. The training sets are used to train the softmax models and interpolate labels for testing set in softmax and WNLL, respectively. Table 1 lists the classification accuracies of WNLL and softmax on three datasets. For WNLL interpolation, in order to speed up the computation, we only use 15 nearest neighbors to ensure sparsity of the weight matrix, and the 8th neighbor's distance is used to normalize the weight matrix. The nearest neighbors are searched via the approximate nearest neighbor (ANN) algorithm [22]. WNLL outperforms softmax significantly in all three tasks. These results show the potential of using WNLL instead of softmax as the output activation function in DNNs.

Table 1: Accuracies of softmax and WNLL in classifying some classical datasets.

| Dataset | CIFAR10 | MNIST | SVHN |
|---------|---------|-------|------|
| softmax | 39.91% | 92.65% | 24.66% |
| WNLL | 40.73% | 97.74% | 56.17% |

For the deep learning experiments below: We take two passes alternating steps, i.e., $N = 2$ in Algorithm. 1. For the linear activation stage (Stage 1), we train the network for $n = 400$ epochs. For the WNLL stage, we train for $n = 5$ epochs. In the first pass, the initial learning rate is 0.05 and halved after every 50 epochs in training linear activated DNNs, and 0.0005 when training the WNLL activation. The same Nesterov momentum and weight decay as used in [12, 17] are used for CIFAR and SVHN experiments, respectively. In the second pass, the learning rate is set to be one fifth of the corresponding epochs in the first pass. The batch sizes are 128 and 2000 when training softmax/linear and WNLL activated DNNs, respectively. For fair comparison, we train the vanilla DNNs with softmax output activation for 810 epochs with the same optimizers used in WNLL activated ones. All final test errors reported for the WNLL method are done using WNLL activations for prediction on the test set. In the rest of this section, we show that the proposed framework resolves the issue of lacking big training data and boosts the generalization accuracies of DNNs via numerical results on CIFAR10/CIFAR100. The numerical results on SVHN are provided in the appendix.

### 4.1 Resolving the Challenge of Insufficient Training Data

When we do not have sufficient training data, the generalization accuracy typically degrades as the network goes deeper, as illustrated in Fig.3. The WNLL activated DNNs, with its superior regularization of the parameters and perturbation on bad local minima, are able to overcome this degradation. The left and right panels plot the cases when the first 1000 and 10000 data in the training set of CIFAR10 are used to train the vanilla and WNLL DNNs. As shown in Fig. 3, by using WNLL activation, the generalization error rates decay consistently as the network goes deeper, in contrast to the degradation for vanilla DNNs. The generalization accuracy between the vanilla and WNLL DNNs can differ up to 10 percent within our testing regime.

Figure.4 plots the evolution of generalization accuracy during training. We compute the test accuracy per epoch. Panels (a) and (b) plot the test accuracies for ResNet50 with softmax and WNLL activations (1-400 and 406-805 epochs corresponds to linear activation), respectively, with only the first 1000 examples as training data from CIFAR10. Charts (c) and (d) are the corresponding plots with 10000 training instances, using a pre-activated ResNet50. After around 300 epochs, the accuracies of the vanilla DNNs plateau and cannot improve any more. In comparison, the test accuracy for WNLL jumps at the beginning of Stage 2 in first pass; during Stage 1 of the second pass, even though initially there is an accuracy reduction, the accuracy continues to climb and eventually surpasses that of the WNLL activation in Stage 2 of first pass. The jumps in accuracy at epoch 400 and 800 are due to

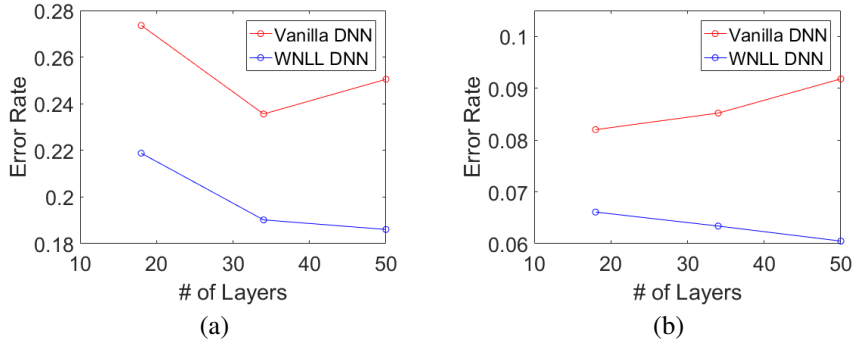

<div align="center">(a)             (b)</div>

Figure 3: Resolving the degradation problem of vanilla DNNs by WNLL activation. Panels (a) and (b) plot the generation errors when 1000 and 10000 training data are used to train the vanilla and the WNLL activated DNNs, respectively. In each plot, we test three different networks: PreActResNet18, PreActResNet34, and PreActResNet50. All tests are done on the CIFAR10 dataset.

switching from linear activation to WNLL for predictions on the test set. The initial decay when alternating back to softmax is caused partially by the final layer $W_L$ not being tuned with respect to the deep features $\tilde{\mathbf{X}}$, and partially due to predictions on the test set being made by softmax instead of WNLL. Nevertheless, the perturbation via the WNLL activation quickly results in the accuracy increasing beyond the linear stage in the previous pass.

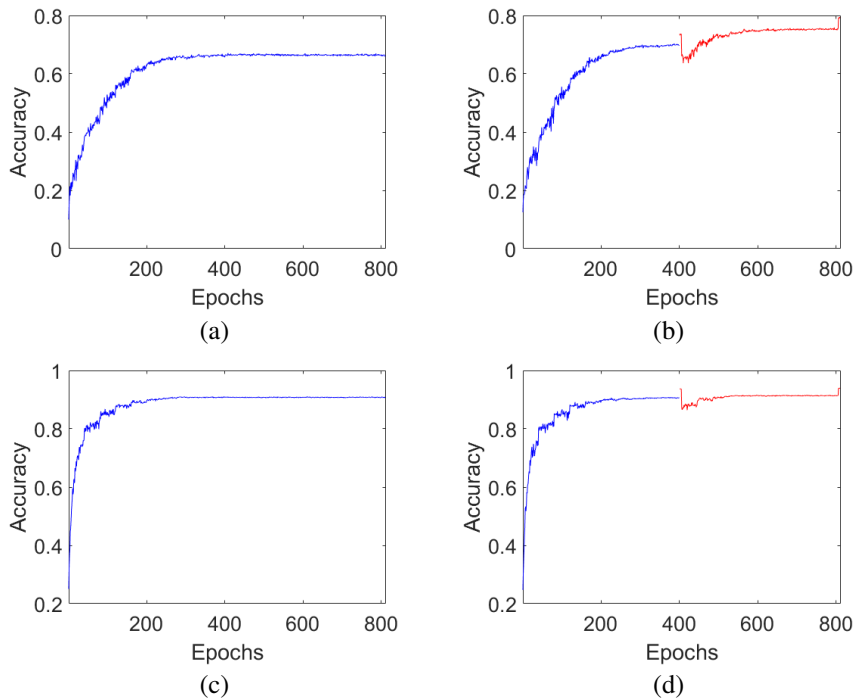

Figure 4: The evolution of the generation accuracies over the training procedure. Charts (a) and (b) are the accuracy plots for ResNet50 with 1000 training data, where (a) and (b) are plots for the epoch v.s. accuracy of the vanilla and the WNLL activated DNNs. Panels (c) and (d) correspond to the case of 10000 training data for PreActResNet50. All tests are done on the CIFAR10 dataset.

## 4.2 Improving Generalization Accuracy

We next show the superiority of WNLL activated DNNs in terms of generalization accuracies when compared to their surrogates with softmax or SVM output activations. Besides ResNets, we also

test the WNLL surrogate on the VGG networks. In table 2, we list the generalization errors for 15 different DNNs from VGG, ResNet, Pre-activated ResNet families on the entire, first 10000 and first 1000 instances of the CIFAR10 training set. We observe that WNLL in general improves more for ResNets and pre-activated ResNets, with less but still significant improvements for the VGGs. Except for VGGs, we can achieve relatively 20% to 30% testing error rate reduction across all neural nets. All results presented here and in the rest of this paper are the median of 5 independent trials. We also compare with SVM as an alternative output activation, and observe that the results are still inferior to WNLL. Note that the bigger batch-size is to ensure the interpolation quality of WNLL. A reasonable concern is that the performance increase comes from the variance reduction due to increasing the batch size. However, experiments done with a batch size of 2000 for vanilla networks actually deteriorates the test accuracy.

Table 2: Generalization error rates over the test set of vanilla DNNs, SVM and WNLL activated ones trained over the entire, the first 10000, and the first 1000 instances of training set of CIFAR10. (Median of 5 independent trials)

| Network | Whole | | | 10000 | | 1000 | |
|---|---|---|---|---|---|---|---|
| | **Vanilla** | **WNLL** | **SVM** | **Vanilla** | **WNLL** | **Vanilla** | **WNLL** |
| VGG11 | 9.23% | **7.35%** | 9.28% | 10.37% | **8.88%** | 26.75% | **24.10%** |
| VGG13 | 6.66% | **5.58%** | 7.47% | 9.12% | **7.64%** | 24.85% | **22.56%** |
| VGG16 | 6.72% | **5.69%** | 7.29% | 9.01% | **7.54%** | 25.41% | **22.23%** |
| VGG19 | 6.95% | **5.92%** | 7.99% | 9.62% | **8.09%** | 25.70% | **22.87%** |
| ResNet20 | 9.06% (8.75%[12]) | **7.09%** | 9.60% | 12.83% | **9.96%** | 34.90% | **29.91%** |
| ResNet32 | 7.99% (7.51%[12]) | **5.95%** | 8.73% | 11.18% | **8.15%** | 33.41% | **28.78%** |
| ResNet44 | 7.31% (7.17%[12]) | **5.70%** | 8.67% | 10.66% | **7.96%** | 34.58% | **27.94%** |
| ResNet56 | 7.24% (6.97%[12]) | **5.61%** | 8.58% | 9.83% | **7.61%** | 37.83% | **28.18%** |
| ResNet110 | 6.41% (6.43%[12]) | **4.98%** | 8.06% | 8.91% | **7.13%** | 42.94% | **28.29%** |
| ResNet18 | 6.16% | **4.65%** | 6.00% | 8.26% | **6.29%** | 27.02% | **22.48%** |
| ResNet34 | 5.93% | **4.26%** | 6.32% | 8.31% | **6.11%** | 26.47% | **20.27%** |
| ResNet50 | 6.24% | **4.17%** | 6.63% | 9.64% | **6.49%** | 29.69% | **20.19%** |
| PreActResNet18 | 6.21% | **4.74%** | 6.38% | 8.20% | **6.61%** | 27.36% | **21.88%** |
| PreActResNet34 | 6.08% | **4.40%** | 5.88% | 8.52% | **6.34%** | 23.56% | **19.02%** |
| PreActResNet50 | 6.05% | **4.27%** | 5.91% | 9.18% | **6.05%** | 25.05% | **18.61%** |

Tables 2 and 3 list the error rates of 15 different vanilla networks and WNLL activated networks on CIFAR10 and CIFAR100 datasets. On CIFAR10, WNLL activated DNNs outperforms the vanilla ones with around 1.5% to 2.0% absolute, or 20% to 30% relative error rate reduction. The improvements on CIFAR100 are more significant. We independently ran the vanilla DNNs on both datasets, and our results are consistent with the original reports and other researchers' reproductions [12, 13, 16]. We provide experimental results of DNNs' performance on SVHN data in the appendix. Interestingly, the improvement are more significant on harder tasks, suggesting potential for our methods to succeed on other tasks/datasets. For example, reducing the sizes of DNNs is an important direction to make the DNNs applicable for generalize purposes, e.g., auto-drive, mobile intelligence, etc. So far the most successful attempt is DNNs weights quantization[2]. Our approach is a new direction for reducing the size of the model: to achieve the same level of accuracy, compared to the vanilla networks, our model's size can be much smaller.

## 5 Concluding Remarks

We are motivated by ideas from manifold interpolation and the connection between ResNets and control problems of transport equations. We propose to replace the classical output activation function, i.e., softmax, by a harmonic extension type of interpolating function. This simple surgery enables the deep neural nets (DNNs) to make sufficient use of the manifold information of data. An end-to-end greedy style, multi-stage training algorithm is proposed to train this novel output layer. On one hand, our new framework resolves the degradation problem caused by insufficient data; on the other hand, it boosts the generalization accuracy significantly compared to the baseline. This improvement is consistent across networks of different types and different number of layers. The increase in

Table 3: Error rates of the vanilla DNNs v.s. the WNLL activated DNNs over the whole CIFAR100 dataset. (Median of 5 independent trials)

| Network | Vanilla DNNs | WNLL DNNs | Network | Vanilla DNNs | WNLL DNNs |
|---|---|---|---|---|---|
| VGG11 | 32.68% | **28.80%** | ResNet110 | 28.86% | **23.74%** |
| VGG13 | 29.03% | **25.21%** | ResNet18 | 27.57% | **22.89%** |
| VGG16 | 28.59% | **25.72%** | ResNet34 | 25.55% | **20.78%** |
| VGG19 | 28.55% | **25.07%** | ResNet50 | 25.09% | **20.45%** |
| ResNet20 | 35.79% | **31.53%** | PreActResNet18 | 28.62% | **23.45%** |
| ResNet32 | 32.01% | **28.04%** | PreActResNet34 | 26.84% | **21.97%** |
| ResNet44 | 31.07% | **26.32%** | PreActResNet50 | 25.95% | **21.51%** |
| ResNet56 | 30.03% | **25.36%** | | | |

generalization accuracy could also be used to train smaller models with the same accuracy, which has great potential for the mobile device applications.

## 5.1 Limitation and Future Work

There are several limitations of our framework to improve which we wish to remove. Currently, the manifold interpolation step is still a computational bottleneck in both speed and memory. During the interpolation, in order to make the interpolation valid, the batch size needs to be quasilinear with respect to the number of classes. tTis pose memory challenges for the ImageNet dataset [7]. Another important issue is the approximation of the gradient of the WNLL activation function. Linear function is one option but it is far from optimal. We believe a better harmonic function approximation can further lift the model's performance.

Due to the robustness and generalization capabilities shown by our experiments, we conjecture that by using the interpolation function as output activation, neural nets can become more stable to perturbations and adversarial attacks [25]. The reason for this stability conjecture is because our framework combines both learned decision boundary and nearest neighbor information for classification.

## Acknowledgments

This material is based on research sponsored by the Air Force Research Laboratory and DARPA under agreement number FA8750-18-2-0066. And by the U.S. Department of Energy, Office of Science and by National Science Foundation, under Grant Numbers DOE-SC0013838 and DMS-1554564, (STROBE). And by the NSF DMS-1737770 and the Simons foundation. The U.S. Government is authorized to reproduce and distribute reprints for Governmental purposes notwithstanding any copyright notation thereon.

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
