[Supplementary Material]

# Supplementary material:
## *Deep Neural Nets with Interpolating Function as Output Activation*

**Bao Wang**
Department of Mathematics
University of California, Los Angeles
wangbaonj@gmail.com

Xiyang Luo
Department of Mathematics
University of California, Los Angeles
xylmath@gmail.com

Zhen Li
Department of Mathematics
HKUST, Hong Kong
lishen03@gmail.com

Wei Zhu
Department of Mathematics
Duke University
zhu@math.duke.edu

Zuoqiang Shi
Department of Mathematics
Tsinghua University
zqshi@mail.tsinghua.edu.cn

Stanley J. Osher
Department of Mathematics
University of California, Los Angeles
sjo@math.ucla.edu

## A   Proof of Theorem 1

*Proof.* Let $X_i, i = 1, 2, \cdots, N$, be the number of additional data needed to obtain the $i$-type after $(i - 1)$ distinct types have been sampled. The total number of instances needed is:

$$X = X_1 + X_2 + \cdots + X_N = \sum_{i=1}^{N} X_i.$$

For any $i$, $i - 1$ distinct types of instances have already been sampled. It follows that the probability of a new instance being of a different type is $1 - \frac{i-1}{N} = \frac{N-i+1}{N}$. Essentially, to obtain the $i$-th distinct type, the random variable $X$ follows a geometric distribution with $p = \frac{N-i+1}{N}$ and $E[X_i] = \frac{N}{N-i+1}$. Thus, we have

$$E[X] = \sum_{i=1}^{N} E[X_i] = \sum_{i=1}^{N} \frac{N}{N - i + 1}.$$

Asymptotically, $E[X] \approx N \ln N$ for sufficiently large $N$.  □

## B   Results on SVHN data

For the SVHN recognition task, we simply test the performance when the full training data are used. Here we only test the performance of the ResNets and pre-activated ResNets. There is a relative 7%-10% error rate reduction for all these DNNs.

Table 1: Error rates of the vanilla DNNs v.s. the WNLL activated DNNs over the whole SVHN dataset. (Median of 5 independent trials)

| Network | Vanilla DNNs | WNLL DNNs |
|---------|--------------|-----------|
| ResNet20 | 3.76% | **3.44%** |
| ResNet32 | 3.28% | **2.96%** |
| ResNet44 | 2.84% | **2.56%** |
| ResNet56 | 2.64% | **2.32%** |
| ResNet110 | 2.55% | **2.26%** |
| ResNet18 | 3.96% | **3.65%** |
| ResNet34 | 3.81% | **3.54%** |
| PreActResNet18 | 4.03% | **3.70%** |
| PreActResNet34 | 3.66% | **3.32%** |