[Reviews · NeurIPS 2018]

Reviewer 1



This paper develops a new data-dependent output activation function base on interpolation function. It is a nonparametric model based on a subset of training data. The activation function is defined in an implicit manner by solving a set of linear equations. Therefore, it cannot be solved directly by backpropagation. Instead it proposes an auxiliary network with linear output to approximate the gradient. Experiment results show that it can improve the generalization ability on a variety of networks. And it also improves the performance and robustness when the data size is small. The strength of the paper is that the proposed algorithm consistently improves performance by combining nonparametric method with DNN. More importantly, it improves the robustness of DNN for small data case. On the other hand, the weakness of the paper is that the complexity of using nonparametric approach is usually high and it requires storage of the template samples. It seems that the algorithm need to store a lot of template samples in experiments (half of the training set)? It is not clear how the number of required template samples scale with problem size. Furthermore, the argument in lines 96-102 for developing the approximate approach for training the model is a bit heuristic. Comments: *If we are using so many template data samples, why can’t we learn a parametric interpolation function by optimizing the objective function (1)? *Figure 1(b) is not correct. The groundtruth label Y should not be used as input to the model during testing time. *In line 75, “is need to be” should be “need to be”. *In the equation above line 93, it is not explained how \hat{X}^{te} is obtained. From Algorithm 1, it can be conjectured that it is obtained by passing X^{te} through DNN and buffer layer. It should be clarified after this equation in the main text. *The discussion in Section 3 is not clear. And the discussion about (5) is unclear and cannot be understood. *In the caption of Figure 3, “plot the generation” should be “plot the generalization”. *In line 179, “(1-400 and 406-805 epochs corresponds to linear activation)”. I think 406-805 should be the part after switching to WNLL? *In lin 199, why mean is used? Why not use mean and standard deviation? *In line 232, “tTis” should be “This”.

Reviewer 2



This paper proposes a novel interpolating function to replace the output layer of deep neural nets, for example the softmax function. The authors use ideas from manifold learning and study a weighted nonlocal Laplacian layer. This results in DNNs with better generalization and more robust for problems with a small number of training examples. For experiments, WNLL is tested on image recognition tasks, such as CIFAR, MNIST, SVHN. Impressive test error reduction results are reported. Pros: This paper is clearly written and easy to follow. The authors did an especially good job on literature reviews. Cons: While the achieved error reduction results on MNIST and SVHN are quite impressive, stronger baselines with more supervision do exist and can do a better job. Is it possible for the authors to apply the same algorithm on a standard transfer learning task? Furthermore, the proposed neural network layer seems general enough for a wide range of applications, it would be of interest to a greater audience if the authors also experiment with, e.g. NLP and speech tasks. As the authors also mentioned in Section 5.1, i.e. limitation and future work, the proposed algorithm poses memory challenges for the ImageNet dataset. I wonder whether the authors could provide specific quantitative results of train/test speed and memory comparison for w/ and w/o WNLL layer. The paper has several typos. For example, the authors mixed up x and y in several places. Line 232, "tTis" should be "this". Reference [12] and [13] are duplicated.

Reviewer 3



In this paper, the authors propose to replace the softmax output activation with a surrogate activation based on weighted nonlocal Laplacian (WNLL) interpolating function. The overall results seem encouraging. The approach shows superiority of generalization accuracy on a wide variety of networks, and reduces CIFAR10 error rates by more than 1.5% in most architectures. Cons: The author didn’t provide run time analysis using WNLL vs the vanilla softmax activation. What’s the extra computational overhead caused by WNLL? In particular, solving the linear system in Equation (2) or Equation (3) can be quadratic, which inevitably increases the time and memory cost. As the authors mention, this also causes the training difficulty for larger dataset such as ImageNet, which raises the concern of its practical usefulness. Minors: There are a few typos throughout the paper. For example, in Section 5.1, “tTis” should be “This”. Also in Figure 1(b), the testing time input should be X solely without the need for Y.